# Heated Humidified High-Flow Nasal Cannula in Children: State of the Art

**DOI:** 10.3390/biomedicines10102353

**Published:** 2022-09-21

**Authors:** Annamaria Venanzi, Paola Di Filippo, Chiara Santagata, Sabrina Di Pillo, Francesco Chiarelli, Marina Attanasi

**Affiliations:** Department of Pediatrics, University of Chieti, 66100 Chieti, Italy

**Keywords:** heated humidified high-flow, children, respiratory failure, oxygen therapy, bronchiolitis

## Abstract

High-flow nasal cannula (HFNC) therapy is a non-invasive ventilatory support that has gained interest over the last ten years as a valid alternative to nasal continuous positive airway pressure (nCPAP) in children with respiratory failure. Its safety, availability, tolerability, and easy management have resulted its increasing usage, even outside intensive care units. Despite its wide use in daily clinical practice, there is still a lack of guidelines to standardize the use of HFNC. The aim of this review is to summarize current knowledge about the mechanisms of action, safety, clinical effects, and tolerance of HFNC in children, and to propose a clinical practices algorithm for children with respiratory failure.

## 1. Introduction

Respiratory diseases are the most frequent cause of hospitalization in pediatric intensive care units (PICUs), of which 15% may require either invasive mechanical ventilation or non-invasive positive pressure ventilation [1].

High-flow nasal cannula (HFNC) is an oxygen therapy that has received growing attention in recent years as an alternative to continuous positive airway pressure (CPAP) for pediatric patients with respiratory failure [2,3]. HFNC was first indicated to treat apnea in premature infants [4], but its use subsequently expanded in recent years to become a popular mode of pediatric respiratory support. The reasons for its increasing application are its tolerability, safety, and easy-to-use devices, as compared to CPAP or other noninvasive ventilation (NIV) devices [5]. HFNC’s ease of use has resulted in its widespread implementation in settings beyond PICUs, such as in emergency departments and in inpatient pediatric wards. Despite the positive feedback from physicians, development of clinical guidelines surrounding the use of HFNC is still lacking.

The methodological heterogeneity of the studies characterized by different settings for vital parameters, definitions of therapeutic failure and environments (i.e., emergency departments vs. general wards vs. intensive care), and the lack of randomized clinical trials (RCTs) in children, make it difficult to implement internationally recognized guidelines.

The aim of this review is to summarize the most recent scientific evidence on the efficacy, indications, and mechanism of action of HFNC, and to propose a possible algorithm for its use in clinical practice with children.

## 2. Device and Mechanism of Action

A typical HFNC system consists of a flow generator, an active heated humidifier, a single-limb heated circuit, and a nasal cannula [6].

### 2.1. Flow Generator

There are three types of flow generators: air-oxygen blenders, built-in flow generators, and entrainment systems. Each can provide a flow up to 60 L/min, containing an oxygen concentration from 21% to 100% [2]. The air-oxygen blender consists of a mechanical air-oxygen blender that is connected directly from a wall supply with the source of air and oxygen, with a flow meter that allows a stable delivery of both inspiratory fraction of oxygen (FiO_2_) and gas flow. The built-in flow generators operate through a turbine that is capable of generating high flows, beginning from the ambient air, without an external source of gas. It guarantees delivery of a fixed FiO_2_ even if the flow changes, because there is an internal oxygen sensor and a valve which maintains the desired FiO_2_. However, high oxygen concentrations cannot be supplied with this system. The third type of flow generator uses an air entrainment system to generate air flow, which can be used to administer high concentrations of oxygen.

### 2.2. Humidifying Devices

It is well known that the use of dry gas has a harmful impact on the respiratory system, leading to mucociliary malfunction, mucus obstruction, mucosa dryness, and ulceration; its use can also induce reactive bronchostruction [7]. For this reason, the use of HFNC requires that heated and humidified air and gas should be administered. Gas conditioning confers several beneficial effects to the patient, such as improving comfort, optimizing mucociliary clearance, and reducing the metabolic effort of the body for gas conditioning. By passing the mixed gas through a narrow bundle of micropored tubes, optimal humidification can be achieved. 

### 2.3. Inspiratory Limb

The main disadvantage of using humidified gas is that part of the humidity is lost as condensation. Inappropriate humidity may negatively impact clinical outcomes by damaging airway mucosa, prolonging the mechanical ventilation, or increasing respiratory effort [8]. Therefore, high-flow devices can have different types of inspiratory circuits, with some of them being equipped with a heating wire embedded in the circuit wall in order to keep the wall temperature sufficiently high to avoid condensation. 

### 2.4. Interface

The circuit is connected to a nasal cannula that is available in different sizes to fit the patient’s nostrils. An important precaution to be adopted is the choice of the cannula based on age and weight. Manufacturers recommend that a cannula should not have a diameter greater than fifty percent of the patient’s nostril, in order to avoid a significant increase in airway pressure, with a subsequent risk of air leakage [9].

## 3. Clinical Indications

### 3.1. Acute Bronchiolitis

Bronchiolitis is an infection that affects the lower respiratory tract of infants and children, causing hospitalization in 10% of cases [10], and requiring ventilatory support for respiratory failure and apnea.

Bronchiolitis is the main indication for HFNC in children with moderate to severe respiratory distress. Franklin et al. [11] showed in an RCT that included 1472 infants < 12 months of age with bronchiolitis, that there was a significantly lower failure rate in children treated with HFNC than in children treated with conventional low-flow oxygen therapy (COT) (12% and 23%, respectively). However, no significant differences were observed in the length of hospital stay or in the duration of oxygen therapy between the two groups. The recent pediatric studies regarding the use of HFNC in children with bronchiolitis are summarized in Table 1.

Lin et al. [12] carried out a systematic review and meta-analysis that included 2121 children with bronchiolitis; the objective was to compare the efficacy of HFNC with both COT and CPAP. The authors found no evident difference in the length of stay, in the duration of oxygen supplementation, or in transfers to PICU, among the three treatment groups. A significant reduction in the incidence of treatment failure was observed in the HFNC group compared with the COT group, although there was a significant increase in the incidence of treatment failure when the HFNC group was compared with the nasal CPAP group.

More recently, in a systematic review and meta-analysis, including children up to 24 months of age with bronchiolitis, Dafydd et al. [13] showed that HFNC was superior to COT in terms of treatment failure rates, and was not inferior to CPAP in terms of therapeutic failure and intubation rates. 

The aforementioned systematic review also included a study by Kepreotes et al. [14] that showed that HFNC is used as a rescue therapy for patients failing on COT. In this latter study, 202 children with bronchiolitis were randomly assigned to either HFNC (101 children) or COT (101 children). The authors observed that HFNC did not significantly reduce the time of oxygen therapy when compared with COT. However, 61% of children who experienced the failure in treatment with low-flow oxygen therapy were rescued using HFNC. These data suggested that the early use of HFNC was not able to modify the natural course in children with moderate bronchiolitis, although it took on a role as a rescue therapy to reduce the PICU admissions. 

Studies comparing HFNC with CPAP have led to conflicting results. A recent meta-analysis involved three RCTs that included a total of 213 infants of ages < 24 months [15]. Two studies showed similar failure rates between the HFNC and CPAP patient groups [15,16]. However, the third study which comprised a larger cohort of patients than the aforementioned ones, showed a statistically higher failure rate in the HFNC-treated group [17].

**Table 1 biomedicines-10-02353-t001:** Use of HFNC in children with bronchiolitis.

Reference	Study Design	Study Population	Comparison	Main Findings
Franklin et al. [11]	RCT	1472 patients < 12 months with bronchiolitis	HFNC vs. COT	Lower treatment failure rate in children treated with HFNC No difference in the length of hospital stay
Kepreotes et al. [14]	RCT	202 patients < 24 months with moderate bronchiolitis	HFNC vs. COT	Similar duration of oxygen therapy in the two groups 61% of children who experienced failure in treatment with COT were rescued with HFNC
Lin et al. [12]	Systematic review	2121 patients with bronchiolitis	HFNC vs. other oxygen therapies (COT, CPAP)	Lower treatment failure in HFNC group than COT Higher treatment failure in HFNC if compared with nasal CPAP No differences in length of stay, duration of oxygen supplementation, or transfer to PICU among the three groups
Dafydd et al. [13]	Systematic review and meta-analysis	1159 children up to 24 months of age with bronchiolitis	HFNC vs. other oxygen therapies (COT, nCPAP)	Lower treatment failure in HFNC then COT Similar therapeutic failure and intubation rates with HFNC and nCPAP
Moreel et al. [15]	Meta-analysis	213 infants < 24 months	HFNC vs. nCPAP	Similar failure rates with HFNC and CPAP in two RCTs A third RCT comprising a larger cohort of patients showed a statistically higher failure rate in the HFNC-treated group

COT, conventional oxygen therapy; nCPAP, nasal continuous positive airway pressure; HFNC, high flow nasal cannula; PICU, pediatric intensive care unit; RCT, randomized controlled trial.

Further studies are necessary to focus on children with bronchiolitis who may better benefit from HFNC, and to identify the ideal cohort of patients who yield the best findings when using HFNC versus COT. It is also important to define the criteria for treatment failure, in order to standardize the study results and maximize the use of HFNC. 

### 3.2. Asthma

Asthma is the most common obstructive respiratory disease in children, and is a frequent cause of visits to emergency and hospital admission [18]. An acute episode of asthma is potentially life-threatening; treatment of an asthma exacerbation requires the use of systemic bronchodilators and inhaled corticosteroids. The most severe asthma exacerbations could benefit from the use of COT. When there is no response to COT, it is possible to use CPAP as rescue oxygen therapy prior to the intubation. The use of CPAP requires careful monitoring for potential complications, such as gastric distension, which increase the risk of vomiting and inhalation [19].

It has been demonstrated that inhalation of cold and dry gas can induce bronchoconstriction and reduce ciliary activity in the airway, thereby worsening an episode of asthma exacerbation [7]. Two retrospective studies showed that the use of HFNC could improve heart rate, respiratory rate, oxygenation, pH, and partial pressure of carbon dioxide (pCO_2_) [20,21]. Baudin et al. [20] showed that in children with severe asthma treated with HFNC, heart and respiratory rates decreased significantly, and the blood gas improved in the first 24 h. These data were confirmed in a study by Martinez et al. [21] that included children aged 4–5 years with moderate-to-severe asthma exacerbation. It showed that patients treated with HFNC had less PICU admissions compared with those treated using an initial flow of < 15 L/min, and that high flow was well tolerated by children and did not require sedation.

Ballestero et al. [22] carried out a study of 62 children aged 1 to 14 years with asthma exacerbation who were randomly assigned to either HFNC or COT. The authors showed that 53% of the children treated with HFNC had improved respiratory dynamics after two hours of treatment, compared with 28% of children treated with COT [22]. 

Table 2 summarizes the literature review regarding the use of HFNC in children with asthma.

Although the current scientific evidence is limited, the use of HFNC appears to play an important role in the treatment of children with acute asthma exacerbation, especially in preventing the escalation of ventilatory support and intubation. Further well-conducted RCTs are necessary to understand what type of patient could better benefit by using high flows during an asthma exacerbation.

### 3.3. Congenital Heart Diseases

HFNC is an effective respiratory support for patients with precarious hemodynamic balance. In an RCT that included pediatric patients with congenital heart disease, it was shown that the use of HFNC reduced desaturations, risks of escalation to NIV, and hypercapnia, while preserving the hemodynamic balance [23].

It is well known that positive end-expiratory pressure (PEEP) obstructs the venous return to the heart by causing an increase in central venous pressure [24]. Thus, the effectiveness of HFNC is evidenced by the lower PEEP values generated in contrast to CPAP, without significantly modifying the central venous pressure [25]. 

Furthermore, Shioji et al. [26] carried out a retrospective study which compared HFNC and NIV for the treatment of acute respiratory failure after cardiac surgery in children with congenital heart disease. There was a lower reintubation rate within 28 days (3% vs. 26%; *p* = 0.04), and a shorter PICU stay (10 vs. 17 days; *p* = 0.009) in the HFNC group than in the NIV group [26]. 

### 3.4. Obstructive Sleep Apnea

Obstructive sleep apnea (OSAS) is another important field of application of HFNC therapy. The current treatment options for OSAS include adenotonsillectomy and nasal-CPAP, which prevent airway collapse.

In view of the risk of facial injury due to nasal CPAP devices, the use of HFNC therapy has been proposed.

In 2009, McGinley et al. showed that the reduction in apnea-hypopnea index (AHI) in 12 children with a mean age of 10 years with mild-to-severe OSAS was similar between HFNC and CPAP [27].

More recently, two studies [28,29] carried out in children with moderate-to-severe OSAS found that HFNC reduced nocturnal respiratory events and improved oxygen saturation (SpO_2_). Ignatiuk et al. [28] observed a significantly reduced obstructive AHI (28.9 [17.6, 40.2] vs. 2.6 events/h [1.1, 4.0]; *p* < 0.001) in 22 children who underwent HFNC because of poor surgical candidacy, residual OSAS after surgery, and CPAP intolerance. Hawkins et al. [29] found in 10 school-aged subjects treated with HFNC a reduced median obstructive AHI (11.1 [8.7–18.8] vs. 2.1 events/h [1.7–2.2]; *p* = 0.002); an increased oxygen saturation mean (91.3% [89.6–93.5%] vs. 94.9% [92.4–96.0%]; *p* < 0.002); a decreased SpO_2_ desaturation index (19.2 [12.7–25.8] vs. 6.4 events/h [4.7–10.7]; *p* = 0.013); and a reduced heart rate (88 [86–91] vs. 74 bpm [67–81]; *p* = 0.004). In a retrospective report of five patients with OSAS, the authors found clinical improvement and a decreased AHI, with less chronic CO_2_ retention after treatment with HFNC [30].

Despite this limited evidence, HFNC might be considered as a rescue option in children with OSAS and CPAP intolerance. Further studies comparing therapy with HFNC and CPAP are necessary in order to define a standardized protocol for the use of HFNC in the treatment of OSAS as well. 

### 3.5. Pneumonia

Pneumonia is a leading cause of respiratory failure in children [31].

Respiratory failure may be caused by pneumonia, which leads to hypoxemia and/or hypercapnia and increases respiratory effort. Although COT with high-oxygen concentrations improves hypoxemia, the delivery of continuous positive airway pressure improves the respiratory distress and hypercapnia [15]. However, the use of CPAP is limited, since it requires greater technical and clinical skills for its use than COT [17].

Therefore, in recent years the use of HFNC has spread due to its ease of use and good tolerance. HFNC regulates the flow of oxygen and its concentration, delivers humidified oxygen, and creates a CPAP effect [32]. 

Currently, few studies have compared HFNC and CPAP treatments in children with pneumonia. A recent RCT, including 84 children < 2 years of age, evaluated the difference between HFNC and CPAP for the treatment of mild to moderate respiratory failure in children with pneumonia [33]. The need for intubation and PICU transfer occurred in 6 out of 43 children in HFNC group and in 4 out of 41 children in CPAP group (14% vs. 10%; *p* = 0.553). There was no significant difference between the two groups in length of hospital stay (8 [7,8,9] vs. 8 days [7,8,9]; *p* = 0.461) and in duration of non-invasive treatment (2 [2, 3] vs. 3 days [2, 3]; *p* = 0.090). Adverse events that were observed in the HFNC group were lower than those experienced in the CPAP group (5% vs. 27%; *p* < 0.005). Specifically, abdominal distension was less common in the HFNC group compared to the CPAP group (5% vs. 17%; *p* = 0.066), and trauma of the nasal mucosa was less common in the HFNC group compared to the CPAP group (0% vs. 14%; *p* = 0.036). No serious adverse events, such as pneumothorax, cardiorespiratory arrest, and asphyxia, were observed in either group. Additionally, sedative use was less frequent in the HFNC group than in the CPAP group (40% vs. 83%; *p* = 0.000). Recent studies about other clinical indications for the use of HFNC in children are summarized in Table 3.

In 43 adult patients admitted to the intensive care unit with COVID-19 pneumonia who received oxygen via COT or HFNC therapy, a lower short-term mortality (50% vs. 84.2%; *p* = 0.019) and need for intubation (54.2% vs. 84.2%; *p* = 0.037) were found in the HFNC group. [34]. Therefore, the authors suggested that HFNC is a safe and effective alternative treatment for acute hypoxic respiratory failure due to COVID-19 pneumonia [35].

## 4. HFNC Setting: Initiation, Maintenance, and Weaning

Currently, there is no standardized protocol for setting HFNC parameters. 

Based on the characteristics of the patient, three parameters must be adjusted independently: the temperature, the gas flow, and FiO_2_.

-Temperature: between 34–37 °C, with an ideal value of 34 °C for the pediatric patient.-Flow: can be set up to 60 L/min. In the most of pediatric studies, the flow is set up on the basis of body weight (1–2 L/kg/min). In bronchiolitis, a flow of 2 L/kg/min seems to offer maximum efficacy with minimal risk of adverse events.-FiO_2_: set up with the aim of obtaining a saturation of 95–97%.

In order to underline the importance of a shared protocol, a recent study evaluated 584 patients with a median age of 20 months before and after the implementation of a protocol for using HFNC. Two hundred ninety-two patients treated after setting up the protocol had a higher initial flow (14.5 L/min vs. 10.0 L/min; *p* < 0.001), faster weaning (4.1 L/min/h vs. 2.4 L/min/h; *p* < 0.001), a lower failure rate of high-flow therapy (10% vs. 17%; *p* = 0.015), and a shorter hospital stay (5.9 days vs. 6.8 days; *p* = 0.006). Therefore, with the implementation of a protocol that established an increase in the initial flow, a faster weaning and a lower need for escalation to NIV or mechanical ventilation was obtained [35].

The flow-chart for the management of therapy with HFNC is outlined in Figure 1.

After setting the parameters of the device, it is necessary to constantly monitor the vital signs (oxygen saturation and heart rate) every 1–2 h for the first 8 h, and then every 4 h for the next 24 h. After the first 60–90 min, the flow can be increased by 0.2 L/kg/min every 10–20 min, up to a maximum FiO_2_ of 0.5 when there is no clinical improvement. It is also important to try to aspirate the gastric secretions. Other methods of ventilation or transfer to the PICU should be considered when there is no clinical response to HFNC therapy.

Weaning can be considered if respiratory dynamics have improved after 24 h.

When a patient’s condition improves, gradually decreasing the set of parameters of the device is recommended, reducing the flow by 0.2 L/kg/min every 2 h and FiO_2_ by 0.05 every 2 h.

## 5. Advantages of the Use of HFNC 

The use of HFNC has been shown to reduce the respiratory rate and the effort of breathing, while improving alveolar ventilation [36]. 

Several underlying mechanisms are hypothesized for the observed beneficial effects of HFNC in children, although they are not yet fully understood.

The main physiological mechanism responsible for improving respiratory support in HFNC therapy is the reduction in dead space [20,35,37].

HFNC provides a wash-out of the nasopharyngeal dead space, which is the volume of air localized at the level of the proximal third of the respiratory tract. In particular, the nasopharyngeal dead space does not participate in gas exchange during respiration. For the relatively large head size of infants and children, the total dead space is 50% greater than that of adults [38].

Several studies [37,38,39,40,41,42] showed that increasing the dead space wash-out resulted in an improvement in ventilation by favoring the elimination of CO_2_. In addition, the reduction in the prosthetic dead space facilitated the pulmonary gas exchange and decreased the ventilator pressure and volume requirements.

Liew et al. [43] carried out a prospective randomized crossover study with 44 preterm infants who received either HFNC or nasal CPAP. The wash-out of the nasopharyngeal dead space was evaluated by measuring the nasopharyngeal end-expiratory CO_2_ (pEECO_2_). The authors found that increasing the flow from 2 to 8 L/min led to a significant reduction in pEECO_2_ and decreased the respiratory rate; the reason for these results was probably because of the reduction in dead space ventilation. The mean nasal-CPAP pEECO_2_ was higher with CPAP compared with all flow rates of HFNC, although it was only significant at 6–8 L/min (*p* < 0.05).

The wash-out of the nasopharyngeal dead space determines a better gas exchange, and allows for reductions in the pressure and volume parameters of HFNC [25,41,44]. 

Several studies [25,45,46] also demonstrated that HFNC significantly reduced the inspiratory resistance at the upper airways, providing a flow of gas that matched or exceeded the inspiratory rate.

Saslow et al. [45] conducted a study that compared the effort of breathing in premature neonates who were supported with HFNC and nasal-CPAP. The authors showed that the effort of breathing with HFNC between 3 and 5 L/min was equivalent to that with nasal-CPAP at 6 cm H_2_O, although there was a lower esophageal pressure in children who were treated with HFNC instead of CPAP (1.32 ± 0.77 vs. 1.76 ± 1.46 cm H2O; *p* < 0.05). These data suggest that the use of HFNC may lead to a reduction in gastric distension, with a better tolerance in children. 

Although PEEP cannot be measured or controlled as it can with nasal-CPAP because of air leakage around the nostrils and oral breathing, the generation of modest PEEP (2–6 cm H_2_O with a flow of 8–12 L/min) is also possible with HFNC therapy [9,17]. This “PEEP effect” assists the residual functional capacity, prevents collapse of the pharynx, and reduces the effort of breathing.

## 6. Adverse Side Effects and Contraindications

High-flow therapy is associated with several adverse events in the pediatric population, such as nasal irritation, epistaxis, and abdominal distension. The latter symptom can be improved with the placement of a nasogastric tube.

The most feared complication is barotrauma, with air-trapping, pneumothorax, and pneumomediastinum. Those complications seem to be associated with the use of inappropriately sized nasal cannulas [5]. A recent meta-analysis of 8 RCTs that included 2259 subjects (1100 assigned to HFNC, 980 to standard oxygen, and 179 to nasal-CPAP) found no difference in the incidence of air leak or pneumothorax among low-flow oxygen therapy, high-flow oxygen therapy, and CPAP groups [43]. Therefore, the occurrence of barotrauma may be related to underlying pathological conditions rather than to the therapy.

However, HFNC is considered less invasive, better tolerated, and associated with fewer complications than mechanical ventilation [44].

Severe hypoxia, which requires invasive ventilation, and hemodynamic instability, represent contraindications for HFNC use. In addition, HFNC is contraindicated where facial trauma or skull base fractures are the mechanical impediment. The presence of pneumothorax contraindicates the use of high flows, which would increase the air trapped in the subpleural space. Lastly, altered consciousness represents a contraindication, since the use of non-invasive ventilatory methods presupposes the presence of spontaneous breathing.

## 7. Comparison with Other Ventilation Techniques

The use of higher oxygen flows allow differentiation between HFNC and COT, which includes different devices, such as nasal cannulas, with the oxygen delivered at lower flows (at a rate of < 2 L/min in infants and < 6 L/min in children). Low flows also differ from HFNC because they do not require humidification to prevent the drying effect of unhumidified cold oxygen, and discomfort to the nasal mucosa. 

A meta-analysis involving 2259 children with respiratory distress and mild hypoxemia caused by bronchiolitis or pneumonia, showed that HFNC significantly reduced the risk of treatment failure compared with COT [47]. More recently, 563 children aged 0–16 years with acute respiratory failure were randomized to high-flow or low-flow oxygen therapy (1:1 ratio). The treatment failure rate was lower in the children treated with high-flows than those treated with low-flow oxygen therapy (11.7% vs. 18.1%; odds ratio 0.62; 95% CI 0.38–1.00). Sixty percent of children who did not benefit from low-flow oxygen therapy successfully responded to high-flow oxygen therapy. No difference in PICU transfer rate nor in the length of hospital stay was observed [11]. 

Despite these promising findings, 67–87% of children with respiratory distress and hypoxemia respond to low-flow oxygen therapy [47]. Therefore, HFNC is not currently recommended as a first-line therapy in children with respiratory distress and mild hypoxemia, aside from its higher cost and complexity of use compared to low-flow oxygen therapy. 

In a recent retrospective study involving 137 children between the ages of 1 month and 2 years, Habra et al. [48] observed a higher failure rate with HFNC compared to BiPAP or CPAP in children with bronchiolitis in PICUs (50.6% vs. 0% for CPAP vs. 8% for BiPAP, *p*  < 0.01). Among those who failed in the HFNC group, 35 patients (90%) were shifted to another mode of noninvasive respiratory support that was therapeutically effective. 

These findings suggested that HFNC may be considered as a bridging mode of ventilation between COT and CPAP, reducing the need for nasal-CPAP. However, there is a need for well-executed RCTs that compare HFNC with other modalities of non-invasive respiratory support for the treatment of severe bronchiolitis.

Table 4 describes the main characteristics and differences among COT, HFNC, and CPAP.

## 8. Clinical Predictive Scores

Several parameters could be evaluated to predict a good response to HFNC therapy. Patient age and a history of prematurity were not associated with treatment failure [45], although premature birth increases the risk of long-term lung disease [49].

Higher pCO_2_ values at baseline are associated with HFNC treatment failure. The respiratory rate is another pivotal parameter that is important to consider: non-responder patients were less tachypnoic at baseline, and their respiratory rates did not decrease significantly after therapy. Therefore, patients with hypercapnia and lower respiratory rates should be frequently monitored when beginning HFNC.

According to Mayfield et al. [50], the reductions in respiratory and heart rates by approximately 20% from baseline in the first 90 min of treatment could represent a predictive marker of treatment efficacy.

A score often used in clinical practice to establish the need for HFNC, as well as predict the therapeutic response to HFNC, is the Pediatric Early Warning System (PEWS). A retrospective study involving patients up to 17 years of age documented that a higher and worsening PEWS score 90 min after HFNC onset was predictive of treatment failure [47].

The rate-oxygenation index (ROXI) is the fraction of oxygen saturation, with FiO_2_ as the numerator and respiratory rate as the denominator. It was also a valid predictor of the need for invasive mechanical ventilation in patients receiving HFNC [51]. Yildizdas et al. [52] created an equivalent pediatric score using the respiratory rate z-score, called the pediatric rate-oxygenation index (p-ROXI). The authors concluded that ROXI and p-ROXI changes could predict treatment failure at 24 and 48 h after HFNC onset.

Lastly, a universally accepted score is not yet available, but would be useful to standardize clinical practice and reduce heterogeneity among studies.

In general, the responder patients present an improvement in respiratory and heart rates, and in respiratory effort within 60–90 min after beginning HFNC. Conversely, an increased oxygen requirement, stable or worsening respiratory and heart rates, or respiratory effort, presuppose the need for a step-up therapy [37].

## 9. Conclusions

In the literature, current evidence supports the use of HFNC mainly in infants with bronchiolitis. Over the last decade, HFNC therapy has reduced the need for non-invasive and invasive ventilation in these children. However, the cost-benefit assessment indicates that standard oxygen therapy still represents the first-line treatment option. It is reasonable to use HFNC as a second-line treatment in infants with bronchiolitis, using nasal-CPAP in case of HFNC failure.

The ease of use, safety, and availability of HFNC have led to its broader application in other clinical settings and conditions, such as for respiratory failure and distress. Despite its widespread use, shared guidelines are still lacking. The indications for the use of HFNC, the parameters settings, and the treatment response still depend on physician expertise. These factors lead to heterogeneous management in clinical practice and difficulties in comparing related studies. Standardization of HFNC management and the development of predictive scores to identify responder patients are needed as soon as possible.

## Figures and Tables

**Figure 1 biomedicines-10-02353-f001:**
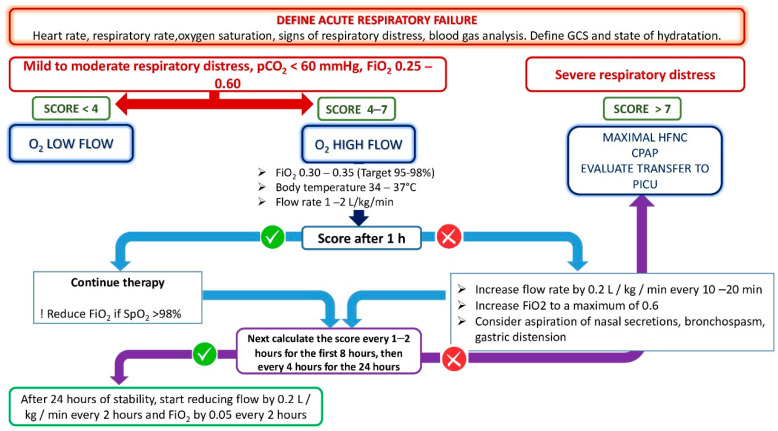
Decisional flow-chart for the management of high-flow oxygen therapy in children. The Pediatric Early Warning System was considered as the score. GCS = Glasgow Coma Scale; pCO_2_ = partial pressure of carbon dioxide; FiO_2_ = fraction of inspired oxygen; O_2_ = oxygen; SpO_2_ = blood oxygen saturation; HFNC = high-flow nasal cannula; CPAP = continuous positive airway pressure; PICU = pediatric intensive care unit.

**Table 2 biomedicines-10-02353-t002:** Use of HFNC in children with asthma.

Reference	Study design	Study Population	Comparison	Main Results
Baudin et al. [20]	Retrospective observational study	73 PICU patients aged 1 to 18 years with severe asthma	HFNC vs. COT	Lower heart, better respiratory rates and blood gas in the first 24 h in the HFNC group
Martinez et al. [21]	Observational study	536 children aged 4–5 years with moderate-to-severe asthma exacerbation	HFNC vs. COT	Lower PICU admissions in the HFNC group
Ballestero et al. [22]	Randomized pilot trial	62 children aged 1 to 14 years with asthma exacerbation	HFNC vs. COT	Improvement in respiratory dynamics after two hours in the HFNC group compared with the COT group

COT, conventional oxygen therapy; HFNC, high flow nasal cannula; PICU, pediatric intensive care unit.

**Table 3 biomedicines-10-02353-t003:** Other clinical indications for HFNC in pediatric patients.

Disease	Reference	Study Design	Study Population	Comparison	Main Results
Congenital heart diseases	Shioji et al. [26]	Retrospective study	35 children with congenital heart disease surgically corrected and acute respiratory failure	HFNC and NIV	Lower intubation rate in the HFNC group Shorter PICU stay in the HFNC group
OSAS	McGinley et al. [27]	Retrospective study	12 children with a mean age of 10 years with mild-to-severe OSAS	HFNC vs. CPAP	Similar reductions in AHI in both groups
	Ignatiuk et al. [28]	Retrospective study	22 children with poor surgical candidacy or residual OSAS after surgery	HFNC vs. no intervention	Significant reduction in AHI with HFNC
	Hawkins et al. [29]	Observational study	10 school-aged patients with OSAS treated with HFNC	HFNC vs. no intervention	Lower median obstructive AHI Decreased oxygen desaturation index Reduced heart rate
Pneumonia	Liu et al. [33]	RCT	84 children < 2 years with pneumonia and mild to moderate respiratory failure	HFNC and CPAP	Similar intubation rate and PICU transfer No significant difference between the two groups in the length of hospital stay Lower adverse events rate (such as abdominal distension, trauma of the nasal mucosa) in the HFNC group

AHI, apnea-hypopnea index; COT, conventional oxygen therapy; CPAP, continuous positive airway pressure; HFNC, high-flow nasal cannula; NIV, noninvasive ventilation; PICU, pediatric intensive care unit; OSAS, obstructive sleep apnea.

**Table 4 biomedicines-10-02353-t004:** Comparison of the characteristics among standard oxygen therapy, high-flow nasal cannula, and continuous positive airway pressure.

	Standard Oxygen Therapy	High-Flow Nasal Cannula	Continuous Positive Airway Pressure
Optimal gas conditioning	/	+++	+
Generation of positive end-expiratory pressure	/	+	+++
Wash-out of nasopharyngeal dead space	/	+++	+
Improvement in mucociliary clearance	/	+++	/
Flow and oxygen concentration setting	/	+++	+++
Reduced breathing effort	/	++	+++
Reduction in upper airway resistance	/	+	+++
Patient’s comfort	++	+	/

/ no effect; + low effect; ++ medium effect; +++ high effect.

## Data Availability

Not applicable.

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
