# Peer review of "Heated Humidified High-Flow Nasal Cannula in Children: State of the Art"

_biomedicines, 2022, doi:10.3390/biomedicines10102353_

Round 1
Reviewer 1 Report
I appreciate the opportunity to review the manuscript for publication in MDPI biomedicines.
The authors review the current knowledge about the HFNC from comprehensive aspects in children and propose an algorithm for the clinical practice of respiratory failure in pediatrics. I feel that the topics are interesting although the manuscript is premature. I have following comments.
Styles of citing reference numbers are not correct. Some of them also do not match, eg. Ref 27.
It is requisite to summarize the literature reviews in “2. clinical indications” for each disease in a set of tables.
A structural burden is that the manuscript is still in immature style. There are numerous sites of grammar errors, erratum in the manuscript, and awkward phrasings that detract from this work.
Author Response
Styles of citing reference numbers are not correct. Some of them also do not match, eg. Ref 27.
We thank the reviewer for his/her comment. We revised the style and the number of citing references according to the authors guide.
It is requisite to summarize the literature reviews in “2. clinical indications” for each disease in a set of tables.
We summarized the literature reviews cited in the paragraph “Clinical indications” in table 1,2 and 3.
A structural burden is that the manuscript is still in immature style. There are numerous sites of grammar errors, erratum in the manuscript, and awkward phrasings that detract from this work.
We thank the reviewer for his/her suggestion. We revised the English language of the manuscript.
Reviewer 2 Report
This review article is important for the algorithm proposed for the clinical practice of children with respiratory failure. There are presented in an adequate manner an up- to - date for the mechanism of actions, safety, clinical effects and tolerance of HFNC in children.
After the analysis of the review manuscript, I have some observations and suggestions:
1. Please insert a synthesis (as a table, usually) with the main characteristics or mechanisms or anything else there are actually described in the flow of the manuscript.
2. It is advisable to put more iconographic elements (figures, tables) to successfully point out the advances or important characteristics of the described device.
3. The references need to be revised according to the authors guide.
Author Response
- Please insert a synthesis (as a table, usually) with the main characteristics or mechanisms or anything else there are actually described in the flow of the manuscript.
We thank the reviewer for his/her relevant comment. We summarized the literature reviews cited in the paragraph “Clinical indications” in table 1,2 and 3.
- It is advisable to put more iconographic elements (figures, tables) to successfully point out the advances or important characteristics of the described device.
We created a table (table 4) to point out the most important mechanisms of the described device.
- The references need to be revised according to the authors guide.
We thank the reviewer for his/her comment. We revised the style and the number of citing references according to the authors guide.
Round 2
Reviewer 1 Report
No additional comments.
Reviewer 2 Report
The authors have correctly answered the suggestions and the manuscript has been improved to be accepted for publication.